# The Anti-Inflammatory Effect of Taurine on Cardiovascular Disease

**DOI:** 10.3390/nu12092847

**Published:** 2020-09-17

**Authors:** Tawar Qaradakhi, Laura Kate Gadanec, Kristen Renee McSweeney, Jemma Rose Abraham, Vasso Apostolopoulos, Anthony Zulli

**Affiliations:** Institute for Health and Sport, Victoria University, Melbourne, VIC 8001, Australia; laura.gadanec@live.vu.edu.au (L.K.G.); kristen.mcsweeney@live.vu.edu.au (K.R.M.); jemma.abraham@live.vu.edu.au (J.R.A.); vasso.apostolopoulos@vu.edu.au (V.A.); anthony.zulli@vu.edu.au (A.Z.)

**Keywords:** taurine, inflammation, cardiovascular, coronary artery disease, renin angiotensin system

## Abstract

Taurine is a non-protein amino acid that is expressed in the majority of animal tissues. With its unique sulfonic acid makeup, taurine influences cellular functions, including osmoregulation, antioxidation, ion movement modulation, and conjugation of bile acids. Taurine exerts anti-inflammatory effects that improve diabetes and has shown benefits to the cardiovascular system, possibly by inhibition of the renin angiotensin system. The beneficial effects of taurine are reviewed.

## 1. Introduction

Taurine (2-aminoethanesulphonic acid) is a sulfur-containing α-amino acid abundantly expressed in animal tissues (including the brain, retina, muscles, and organs throughout the body) [1]. Taurine is present in high amounts in: shellfish (scallops, mussels, clams), red meat, organ meats, chicken, turkey, eggs, and more recently, in energy drinks [1]. Taurine can also be synthesized endogenously from dietary intake of methionine to the homocysteine conversion pathway in humans [2]. Homocysteine is converted into cystathionine, which in turn is converted into cysteine [2]. Molecular oxygen is then incorporated into cysteine to form cysteinesulfinate, which leads to the production of hypotaurine [2]. The oxidation of hypotaurine, via an unknown enzyme, renders taurine [3].

In humans, taurine has shown anti-inflammatory effects, blood pressure regulation, and at high levels, may protect against coronary heart disease (CAD) [4]. Given that taurine has anti-inflammatory properties, it is an important amino acid in controlling cardiovascular disease (CVD), as inflammation has been suggested to be one of the major contributors to type-2 diabetes (a risk factor of CVD) [5,6,7]. Plant-based diets need to be well planned if they are to provide cardiovascular benefits, as there are differences in nutrients between plant-source foods, which may lead to nutrient deficiencies [8]. Considering that meat consumption is a major contributor to global warming [9], certain lifestyle choices which reduce or eliminate animal product intake using renewable food sources are needed to supplement dietary patterns, which may not be adequate. Furthermore, diets that lack adequate taurine levels can adversely impact on the health status of the fetus during pregnancy, which may lead to increased risk of pathologies in adulthood [10]. Seafood is already considered a recommended and essential aspect of a healthy diet globally. However, whether increasing the intake of seafood (relatively high in taurine concentration) to reduce the development of CAD and thus CVD remains an open question.

The taurine content per 100 g in seaweed (>600 mg), oysters (396 mg), and fish (130 mg), compared to beef (43.1 mg), chicken (17.8 mg), pork (61.2 mg), and lamb (43.8 mg) [11,12], clearly suggests that research on the biological effects of relatively high taurine is required to be investigated. Indeed, it has been documented that the weekly seafood consumption in Japan (up to 580 g/week) compared to the Western world (USA: as low as 63 g/week) might be associated with lower CVD and longer life expectancy compared to the Western world [13,14]. The focus of this review is to demonstrate the beneficial effects of taurine on CVD complications and factors that are associated with the risk of CAD (i.e., type-2 diabetes and obesity).

## 2. Taurine Deficiency

Almost five decades ago, it was noted that a diet deficient of taurine led to the development of retinopathy in cats [15]. Subsequently, it was shown that cats and dogs with taurine deficiency developed cardiomyopathy [16,17]. Although the effects of taurine deficiency in humans are largely unknown, it is clear that infants depend on taurine for their neurodevelopment [18]. In fact, it was shown that a low score on the mental development index at 18 months of age was associated with low plasma neonatal taurine concentrations [18]. In addition, children with a defective taurine transporter gene (SLC6A6) experience taurine deficiency, and develop retinopathy and cardiomyopathy similar to that noted in feline and canine studies [19]. Interestingly, maternal taurine deficiency is associated with lower weight and length of newborns at birth, compared with pregnant women with high taurine intake [20]. Barker et al. observed that the increase in size of newborns at birth may be protective against the development of CAD [21], thus suggesting that taurine deficiency in infants may lead to the progression of CAD in adulthood. The mechanism of taurine deficiency leading to the cardiac pathologies was shown in taurine transporter (TauT) gene knockout mice [22,23]. The hearts in TauT knockout mice have impaired mitochondrial complex I, resulting in elevated superoxide production (oxidative stress) within the mitochondria [22,23]. This ultimately causes endoplasmic reticulum stress and apoptosis, thought to be the attribute of cardiac failures [22,23].

## 3. Effects of Taurine on the Cardiovascular System

Populations with higher meat intake compared to high seafood intake have much higher CVD-related death rates [24]. Taurine content in seafood is much higher compared to meat, suggesting that taurine may play a role in preventing cardiovascular complications. Studies have determined a correlation between elevated taurine intake and reduced CVD risks, including CAD incidence [25]. In an animal model, taurine supplementation in rabbits fed an atherogenic diet reduced the elevated CVD risk factor, known as hyperhomocysteinemia, and reduced atherosclerosis present in the left main coronary artery [26], thus further suggesting a beneficial effect of taurine on blood vessels. 

Taurine causes a direct effect on blood vessel function when administered to arteries in vitro [27]. When taurine was administered to thoracic aortae harvested from male Wistar rats, vasorelaxation was induced in a dose-dependent manner via the opening of an unspecified potassium channel [27], possibly the delayed outward K^+^ channel current [1], as impaired vasorelaxation is one of the main factors leading to CVDs [28]. Therefore, taurine-induced vasorelaxation may elicit a beneficial role in the progression of CVD. Taurine predominantly remains as an unbound, free amino acid, as opposed to being metabolized into peptides or proteins [29]. It acts as a zwitterion molecule with a relatively strong hydrophilic nature, which enables taurine to have a role in osmoregulation [29]. As such, other mechanisms of taurine-induced vasorelaxation have been proposed through its osmoregulatory effects [30]. In endothelial cells (the outermost layer of blood vessels regulating vasorelaxation), taurine efflux is involved in regulating volume decrease upon osmotic cell swelling [30]. Taurine uptake is involved in volume increase upon osmotic cell shrinkage, which leads to extracellular and intracellular electrolyte/ionic concentration alterations, causing membrane permeability [30]. Further experiments are required to verify taurine-induced vasorelaxation.

Other studies have highlighted the capacity of taurine to elicit anti-apoptotic effects on cardiomyocytes [31,32]. Cultured neonatal rat cardiomyocytes exposed to ischemic conditions at varying time periods (24 to 72 h) with apoptosis showed the advantageous effects of taurine on the intrinsic apoptosis pathway [31,32]. Untreated cells exposed to ischemic conditions showed evidence of mitochondrial depolarization, cytochrome-c release, activation of the apoptotic pathway, and caspase-3 and caspase-9 expression [32,33]. In taurine-treated cells, there was no indication of attenuation in the mitochondrial membrane potential or a reduction in cytochrome-c release. However, there was evidence of taurine-induced inhibition of caspase-3 and caspase-9 activity and reduced levels of apoptosis over 72 h [32]. These results were attributed to inhibitory effects of taurine on the formation of the Apaf-1/caspase-9 apoptosome [32]. It was also noted that taurine inhibited apoptosis, specifically through Akt-facilitated caspase-9 inactivation [31]. Taurine displays cardioprotective abilities following cardiac injury [34]. Wistar rats that received orally administered taurine (100 mM) 30 days following coronary artery occlusion-induced ischemia had significantly smaller myocardial infarct size, elevated levels of superoxide dismutase, and decreased inflammatory markers of interleukin (IL)-6 and tumor necrosis factor alpha (TNFα), compared to controls [33]. This was suggestive that taurine supplementation may be an effective approach in cardio-protection.

Effects of Taurine on the Renin-Angiotensin System

A regulator of the cardiovascular system is the renin-angiotensin system (RAS), a system exhibited on various tissues and cells, which has been established as a key regulator of the body’s hormonal and fluid balance system. The RAS is usually pharmacologically targeted for the management of heart failure [34,35,36,37], diabetes [38,39], and hypertension/blood pressure control [40,41,42]. The main effector of RAS is angiotensin II (AngII), which is produced by angiotensin-converting enzyme (ACE) within the circulation or in ACE-expressed cells [43,44,45]. AngII stimulates the AngII type 1 receptor (AT1R) and AngII type 2 receptor (AT2R) on blood vessels to cause vasoconstriction and vasodilation, respectively [44,45,46]. Another enzyme of RAS is angiotensin-converting enzyme II (ACE2), which is important for producing the other components of RAS, such as angiotensin (1–7) (Ang 1–7) and alamandine, both of which stimulate the oncogene receptors, Mas R and Mas-related G-coupled protein receptor member D (MrgD R) [46]. Both alamandine and Ang (1–7) promote vasodilation, which is beneficial for blood pressure regulation and ultimately homeostasis of the cardiovascular system [44,47]. However, most of the components of RAS are dysregulated, overexpressed, or under-expressed in CVDs, including diabetes, atherosclerosis (during and after endothelial dysfunction), and hypertension [35].

Evidence suggests that taurine also interacts with the RAS in multiple disease models [48,49]. For example, in diabetic rats, taurine regulates RAS by decreasing the expression of the AT2R in cardiomyocytes [50]. Taurine has also been reported to directly inhibit the actions of AngII in neonatal rat myocyte, and prevent AngII-induced cell proliferation and gene expression (e.g., cellular-fos (oncogene) and cellular-jun(proto-oncogene) genes) in cultured cardiac fibroblasts [51,52,53]. As AngII could trigger the Na^+^/Ca^2+^ exchanger, leading to an influx of intracellular calcium ion ([Ca^2+^]i), it is possible that taurine may regulate this ion exchanger in cardiac myocytes and inhibit AngII effects on cardiac hypertrophy [53]. Interestingly, AngII-incubation followed by short-term taurine (24 h) administration in neonatal rat myocyte decreased [Ca^2+^]i [53]. However, vice-versa incubations had no effect on the AngII-induced increase in [Ca^+^]i [53]. As such, it is not clear whether taurine directly inhibits AngII via AT1R, blocks the conversion of AngI into AngII, or simply influences the signaling cascade of AngII in cardiac hypertrophy. Bkaily et al. have proposed that short-term taurine supplementation may stimulate Ca^2+^-dependent mechanisms, whereby taurine is co-transported with Na^+^ on cardiomyocytes, subsequently leading to an increase of [Na^+^]i and [taurine]i [1]. The increase in [Na^+^]i could signal the Na^+^/Ca^2+^ exchanger, resulting in elevated [Ca^2+^]i [1]. Although short-term taurine supplementation may be beneficial, as cardiomyocytes rely on [Ca^2+^]i for contractility function [1], further studies are required to examine the mechanism of taurine’s effect on AngII in cardiac hypertrophy.

In an atherogenic rabbit model, immunohistochemical analysis of the endothelial layer of rabbit left main artery showed that taurine interacts with RAS by reducing ACE2, AT2R, ACE, and AT1R in atherogenic group [26]. In another study, rats were exposed to mechanical stress to elevate blood pressure and supplemented with 200 mg/kg/day of taurine [49]. It was found that in the control group, ACE was overactivated, contributing to the mechanism of elevated blood pressure [49]. This overactivation of ACE was counterbalanced with ACE2 upregulation after taurine supplementation, and was correlated with a reduction in blood pressure [49]. Therefore, in the early stages of vascular disease, such as hypertension, taurine can interact with the beneficial axis of RAS via upregulating ACE2. Taken together, the studies above describe the beneficial and direct effect of taurine on the cardiovascular system (Figure 1).

## 4. Beneficial Effects of Taurine in Diabetes and Obesity 

Diabetes is a well-established risk factor associated with CAD and is a common comorbidity with CVD. Patients with type-2 diabetes have an elevated proinflammatory profile, which drives the progression of atherosclerosis (i.e., endothelial dysfunction), leading to arterial media/intima thickening and plaque formation [5,7]. The progression of CAD in patients with diabetes accelerates, as a cohort study reported that after a 2.5 year follow-up, patients developed new plaques, angina, and carotid intima/media thickness [6], despite 80% of the patients taking medications. As such, treating hyperglycemia is crucial before the disease accelerates CVD development.

Numerous studies have emerged in recent years determining the beneficial effects of taurine supplementation on reducing the onset of diabetes and its associated complications. In fact, in a study in alloxan-induced type-1 diabetic rat models, taurine supplementation prevented beta cell injury in the pancreatic islets, with an increase in glycogen, insulin, and C-peptide content, and a decrease in fructosamine and glucose in the liver [54]. Similar studies in alloxan-induced diabetic rabbit models also found taurine to prevent hyperglycemia, exerting a normalizing and protective role in insulin-dependent diabetics [55,56]. In addition, L’Amoreaux et al. demonstrated that the uptake of taurine in pancreatic cell lines leads to the alteration of the electrical potential of the beta cells, resulting in decreased intracellular insulin levels [57]. This suggests that taurine may also play a crucial role in regulating insulin release from pancreatic beta cells into the plasma to reduce elevated glucose in healthy people (Figure 1). Further studies are required to determine whether this mechanism may also be a potential therapeutic advance for type-2 diabetes by causing a decrease in plasma glucose levels.

The Otsuka Long-Evans Tokushima fatty (OLETF) diabetic rat is a gold standard type-2 diabetes model [58]. In a study conducted on OLETF rats aged 50 weeks with chronic diabetes, taurine supplementation resulted in decreased resistance to insulin, and decreased serum glucose and lipid concentrations [58]. While hyperglycemia was reduced, the complications that had already arisen due to the condition were not reversed, irrespective of the reduced serum glucose and lipids [58]. Taurine was administered (1.5 g/day for 8 weeks) to overweight non-diabetic males (n = 20) who were genetically predisposed to type-2 diabetes. Researchers found no change in the action and secretion of insulin, and additionally, no effect on the serum lipid levels after 8 weeks of taurine supplementation [59]. Therefore, further research is required in diabetic patients to understand the degree in which taurine exerts cytoprotective properties can contribute to the prevention and treatment of type-2 diabetes in larger randomized population groups. 

Those who are considered overweight or obese are at a higher risk of developing type-2 diabetes and insulin resistance, conditions that are associated with accelerated CAD [60]. Obesity is associated with risk factors for CAD due to hyperglycemia/diabetes, high cholesterol, high blood pressure, and metabolic syndrome. Interestingly, obesity itself is an independent risk factor for CAD [61]. Ventricular hypertrophy, diastolic dysfunction, and aortic stiffness are all pathologies associated with both functional and structural changes to the heart caused by obesity [62]. Vascular atherosclerotic lesions are far more prevalent and severe in patients with higher body mass indexes (BMIs) compared to those with normal BMIs, indicating a link between prevalence of atherosclerosis and obesity [63]. Obese patients present with overexpression of pro-inflammatory cytokines, including TNF-*α* and IL-6, and immune cells (phagocytes and macrophages), resulting in elevated inflammation [64]. Taurine has shown to be beneficial in reducing obesity-related inflammation. Obese women administered taurine for an eight week period had increased adiponectin levels and a reduction in inflammatory biomarkers (including high-sensitivity C-reactive protein) and lipid peroxidation [65]. This highlights a role of taurine supplementation in reducing obesity-related inflammation, which can reduce the development of CAD. A key target to prevent CAD is through weight loss. Taurine deficiency is associated with promoting obesity in both animal and human studies [66], a cycle that is negated by taurine supplementation [67]. It has been suggested that supplementation with taurine or activation of its synthesis may be a potential therapy to reduce obesity. Following this hypothesis, a recent publication in a mouse model of obesity showed that prolonged taurine supplementation resulted in weight loss, suggested to be via adipogenesis inhibition [68]. Given this information, supplementation with taurine may be an option to reduce the risk of CVD and CAD in obese patients through reducing BMI.

In addition to the overproduction of pro-inflammatory cytokines during the progression of atherosclerosis, chronic inflammation is also activated in patients with type-2 diabetes. Biomarkers of pro-inflammatory cytokines, including TNF-α and IL-6, are elevated in type-2 diabetes associated with the acceleration of atherosclerosis progression [69,70]. Targeting the inflammatory cascade during CVD is expanding rapidly, as demonstrated by clinical trials involving anti-inflammatory agents [71]. As such, the effect of taurine in inflammation is discussed in the next section.

## 5. Taurine, the Innate Immune System, and Potential Therapeutics in Cardiovascular Disease

### 5.1. Taurine, Toll-Like Receptors, and Cardiovascular Disease

The innate immune system is an evolutionary preserved host defense system, responsible for first-line protective mechanisms against infiltrating pathogens [72]. Integral to innate immunity is a class of pattern recognition receptors, referred to as toll-like receptors (TLRs) [73]. TLRs are fundamental to innate immunity, as they recognize pathogen-associated molecular patterns (highly conserved, distinct motifs present on pathogens) and respond by exerting robust inflammatory mechanisms in an effort to neutralize and eliminate pathogenic invasion [74,75,76]. Furthermore, TLRs are able to respond to danger-associated molecular patterns (host-derived endogenous alarmins), released by damaged, dying, or necrotic cells, independent of pathogenic infection [77]. The end product of sterile inflammation, either through the myeloid-differentiation factor-88 (MyD88)-dependent pathway (TLR1, 2, 4–10) [78] or the MyD88-independent pathway (TLR3 and 4) [79], is ubiquitous among TLRs [80]. However, they display diversity and specialty, as each TLR is activated by unique and specific ligands [81,82,83]. Chronic dysregulated TLR activation has been reported in a plethora of non-infectious pathologies and autoimmune diseases, including CVD [84,85], atherosclerosis [86,87], and type-2 diabetes [88,89]. As demonstrated by animal TLR knockout models [90,91,92,93,94,95,96] and studies involving TLR deficiency [97,98] and polymorphisms in humans [99,100,101,102,103,104,105,106,107,108,109,110,111], absence or improper function of TLRs can attenuate inflammation, influence disease susceptibility, and restore homeostatic balance. Thus, TLRs have become targets of TLR inhibition therapies [112,113,114,115,116]. This review will discuss the potential inhibitory and immunomodulating abilities of taurine and its metabolites on TLR signal transduction, which may make them suitable anti-inflammatory therapeutics in CVD pathologies.

Taurine and TauT localization have been reported in cells of the innate immune system, including lymphocytes [117], neutrophils [118], phagocytes [119], and leukocytes [120]. While taurine does not directly associate with TLRs, it has been suggested that taurine may (i) act as an antioxidant during inflammation induced by oxidative stress [121,122], (ii) exert cytoprotective [123,124], anti-inflammatory [122], and anti-neoplastic [125] effects, and (iii) maintain homeostasis during acute and chronic inflammation [118]. Furthermore, pre-treatment with taurine in rat models of *Streptococcus uberis* mastitis was able to significantly decrease mRNA expression of TLR2, nuclear factor-kappa beta (NF-Kβ), and NF-Kβ DNA binding activity [126] (Figure 2), reducing inducible nitric oxide (iNOS) and TNFα concentrations. Interestingly, TLR4 mRNA expression was significantly increased by taurine supplementation [126], suggesting an in vivo TLR–taurine immunomodulating relationship. Therefore, further studies into the direct relationship between taurine and TLRs are warranted.

The protective and anti-inflammatory effect of taurine has been observed in rodent models of acute lung injury (induced by bleomycin, endotoxin, and cigarette smoke exposure), central nervous system injury, and chemotherapy toxicities. Rodents treated with taurine prior to and after bleomycin-induced acute lung injury showed an absence of pulmonary fibrosis and reduced inflammation markers, including iNOS, intracellular adhesion molecule, TNFα, NF-Kβ, and interleukin (IL) 2, when compared with the untreated group [127,128,129,130,131]. Similar anti-inflammatory effects of taurine have been replicated in endotoxin [132,133] and cigarette smoke [134] models of acute lung injury. After spinal cord injuries, treatment with taurine was able to reduce inflammation, decrease accumulation of neutrophils and secondary degenerative consequences in gray matter, attenuate motor disturbances, and promote axonal regeneration [135,136]. Comparable protective anti-inflammatory results were also observed in traumatic brain injuries, as treatment with taurine was able to significantly reduce pro-inflammatory cytokines (including IL-1α, -1β, -4, -5, -6, -10, -12p70, -13, -17, TNFα, eotaxin, granulocyte colony-stimulating factor, granulocyte-macrophage colony-stimulating factor, interferon-γ, monocyte chemotactic protein-1, and vascular endothelial growth factor) and alleviate severity of the injury [137,138]. Finally, taurine supplementation has shown to protect cardiac [139], renal [140,141,142], and gonadal [143] tissue from cisplatin-induced toxicities, by reducing inflammation caused by oxidative and endoplasmic reticulum stress. Interestingly, while taurine supplementation shows anti-inflammatory effects, TauT deletion shows increased susceptibility [144] and loss of self-healing [145] to parasitic Malaria infection. This suggests that taurine may play a role in launching innate immune system protective inflammatory responses against pathogen infection, or that taurine may be able to arrest growth and development during sexual and asexual stages [144]. Due to the anti-inflammatory nature of taurine, it may have cardioprotective effects in CVDs, which are described as chronic inflammatory diseases [84,85].

Prolonged taurine deficiency in animal models has resulted in immune system dysfunction and abnormalities. Feline models fed a taurine-free diet developed leukopenia, a shift in polymorphonuclear and mononuclear leukocytes percentages, increased total monoclear leukocyte count, alterations in sedimentation of white blood cells, and augmented concentration of serum γ-globulin [146]. Functional studies performed on polymorphonculear cells demonstrated significantly lower numbers of neutrophils in the respiratory tract and a reduction in phagocytosis of *Staphylococcus epidermidis* in felines fed a taurine-free diet [146]. Furthermore, lymph node and spleen histological examination demonstrated regressed follicular centers with depleted fibroblasts (reticular cells), mature and immature B lymphocytes, and mild extravascular hemolysis [146]. To date, the direct correlation between taurine deficiency and immune consequences in humans is elusive. However, it is suggested that TauT expression may be upregulated during inflammation [147]. Furthermore, the anti-inflammatory properties of taurine may be mediated through regulation of TauT transporter in leukocytes [122]. Following anti-depressant treatment, diminished TauT expression was observed in lymphocytes, resulting in augmented extracellular taurine concentration [122]. The efflux of taurine may account for protection against oxidative stress and inhibition of pro-inflammatory cytokine-mediated damage by enabling an increase in its metabolites, taurine-bromamine (TauBr) and taurine-chloramine (TauCl) [122]. TauCl and TauBr have been shown to modulate the TLR/MyD88/NF-Kβ signal transduction in human and murine macrophages. This will be further investigated in the review, focusing on the potential therapeutic abilities of TauCl and TauBr to target inflammation through TLR pathways. The potential benefits these metabolites may have on inflammation and CVD will also be discussed.

### 5.2. Therapeutic Aspects of Taurine-Bromamine

Taurine is the single most abundantly free amino acid found in leukocyte cytosol (20–50 nM) [148], which exerts antioxidant effects by scavenging reactive oxygen species (ROS), such as hypohalous acids [149]. Hypobromous acid (HOBr) is a ROS generated by activated neutrophils and eosinophils at sites of inflammation through enzymatic activity of neutrophil myeloperoxidase (MPO) [150] and eosinophil peroxidase [151]. Taurine readily reacts with HOBr to produce the less toxic conjugate taurine-bromamine (TauBr) [152]. TauBr protects cells against HOBr damage [150], and has demonstrated antimicrobial [153,154], antibacterial [155], anti-inflammatory [148], and antioxidative [148] effects. TauBr has been suggested as a potential topical treatment for *propionbacterium granulosum* in acne vulgaris, due to its antimicrobial and anti-inflammatory effects. TauBr demonstrated potent antibacterial effects similar to that of hypochlorous acid (HOCl) at low concentrations, while remaining cytoprotective [153,156]. TauBr exerts its anti-inflammatory effects by downregulating TNFα-induced NF-Kβ inflammation by inhibiting oxidation and degradation of nuclear factor of kappa light polypeptide gene enhancer in B-cells inhibitor alpha (IkBα) [157]. Additionally, in vitro treatment of TauBr in immune cells and fibroblasts is able to reduce the production of TNF-a, IL-6, IL-10, and IL-12p40 [155,158], and upregulates heme oxygenase-1 (HO-1) [159]. HO-1 is an enzyme that is able to restore homeostatic antioxidant balance during CVD and atherosclerosis [160]. The ability of TauBr to associate with and modulate TLR expression and activation has yet to be investigated. However, TLRs 1, 2, and 4–10 exert pro-inflammatory modulations through the MyD88-dependnet/NF-Kβ pathway, which utilize IkBα (nuclear factor of kappa light polypeptide gene enhancer in B-cells inhibitor, α)for progression of signal transduction [78,161]. Chronic unregulated activation of TLRs has been demonstrated in vascular pathologies, including CVD [84,85] and atherosclerosis [86,87]. Therefore, as TauBr can increase HO-1 and downregulate NF-Kβ inflammation, its investigation into the ability to attenuate exacerbated and chronic inflammation in CVD and atherosclerosis through TLR signal transduction disruption should be investigated. 

### 5.3. Therapeutic Aspects of Taurine-Chloramine

HOCl is a ROS, secreted by macrophages, neutrophils, and eosinophils during inflammation [162,163,164]. HOCl acts as an endogenously produced antibacterial [165,166], antifungal [167], antiparasitic [118], and antiviral agent [118]. However, excessive and unregulated production leads to oxidative stress [168], inflammation [169], and necrosis [170,171]. Dysregulation of HOCl has been associated with pathologies including atherosclerosis [172,173,174], acute coronary syndrome [175], obesity [176], and type-2 diabetes [177]. Taurine reacts with HOCl to form TauCl [178]. TauCl is a less toxic oxidant that has cytoprotective properties [179], reduces acute [180] and chronic inflammation [181], decreases oxidative stress [182], and is able to cause leukocytes damaged by acute inflammation to undergo apoptosis while preserving healthy cells [178,183,184,185]. Unlike TauBr, TauCl has been shown to inhibit mediators of TLR2, 4, and 9 inflammation. Thus, TLRs could be a potential target of TauCl to attenuate chronic inflammatory pathologies. While TauCl is not an activating ligand of TLRs, TauCl-treated macrophages with lipoarabinomannan and zymosan (TLR2-activating ligands), lipopolysaccharide (TLR4-activating ligand), and CpG (Cytosine phosphate guanine) oligonucleotides (TLR9-activating ligand) demonstrated inhibited production of iNOS, NO, and TNFα in a dose-dependent manner (maximal efficacy was observed at 0.8 mM and resulted in 99% of NO and 48% of TNFα) [186,187]. Furthermore, TauCl was able to suppress upregulation of TNFα and iNOS mRNA [186,187]. Therefore, TauCl could have a potential physiological role during protection against tissue injury produced by TLR inflammation. Additionally, TauCl is able to exert anti-inflammatory effects through modulation of NF-Kβ signal transduction and inflammation. TauCl is able to prevent extracellular regulated kinase 1/2 and NF-Kβ activation [188], and migration of NF-Kβ into the nucleus in TauCl-treated rat macrophages [184] and human synovial fibroblasts [178]. TauCl is also able to prevent degradation of IkBα by modifying the molecular backbone [178,184,189]. Decreased phosphorylation and resistance to degradation of IkBα results in decreased activity of IkB kinase required for NF-Kβ signaling [178,184,189]. Taken together, TauCl may exert its anti-inflammatory effects by indirectly preventing TLR activation, achieved by preventing molecules necessary for TLR/MyD88/NF-Kβ signal transduction and transcription of pro-inflammatory products. Inflammatory contributions by TLRs have been implicated in the pathogenesis and disease development of CVD diseases [87]. TLR2 and 4 have been shown to be atherogenic, while conflicting research regarding TLR9’s involvement in CVD pathologies has been shown in the literature [87]. TLR2-deficient mice susceptible to CVD and in vitro studies using human cells which investigated atherogenicity of TLR have shown that TLR2 plays a role in regulating ROS-induced endothelial and vascular inflammation [190], plaque formation [191], migration of vascular smooth muscle cells from the tunica media to the intima [192], and influences plaque stability [193]. Comparable results have been demonstrated in studies involving TLR4, as TLR4 has been associated with endothelial and vascular inflammation [194,195] and atherosclerotic plaque formation [196], and increases macrophage, foam cell, and lipid content of plaques [196,197,198,199], hypertension [200], plaque rupture [201], and migration of vascular smooth muscle cells [202]. Finally, TLR9 has shown to influence plaque formation and macrophage infiltration [203] and endothelial regeneration [204]. TauCl may provide potential therapeutics in CVDs by targeting TLR2, 4, 9/MyD88/NF-Kβ and dampening the inflammatory response (Figure 2).

TLRs have been shown to play a role in chronic inflammatory diseases, including CVD [84,85] and atherosclerosis [86,87]. Deficiencies and polymorphisms have shown beneficial results in disease severity and susceptibility [97,98,99,100,101,102,103,104,105,106,107,108,109,110,111]. Thus, TLRs have become targets of pharmaceuticals and small molecules treatments in TLR inhibition therapies [112,113,114,115,116]. TLRs exert damaging effects through chronic unregulated inflammation through the TLR/MyD88/NF-Kβ pathway [80]. Taurine and its metabolites (TauBr and TauCl) may represent therapeutic amino acids that can indirectly reduce inflammation caused by TLR activation by disrupting normal signal transduction and transcription of pro-inflammatory molecules. Therefore, taurine, TauBr, and TauCl could have potential for clinical application in CVD by dampening inflammation and restoring homeostatic balance. However, more research is required to delineate the relationship between taurine, TarBr, and TauCl interactions with TLRs. 

## 6. Taurine as a Therapeutic in Cardiovascular Disease: Human Clinical Studies

The effect of taurine in patients with CVD has been assessed in several clinical studies (Table 1). In a randomized single-blind placebo-controlled study, it was noted that supplementation of taurine (500 mg/3 times/day for 2 weeks) in patients with heart failure (HF) due to CAD was able to increase exercise capacity [205]. In fact, exercise in HF patients has been shown to improve circulating angiogenic cells [206]. In addition, taurine supplementation (500 mg/3 times/day for 2 weeks) in patients with HF significantly reduced total cholesterol, triglyceride levels, high-density lipoprotein cholesterol, and inflammatory biomarker hsCRP (high sensitivity C reactive protein) levels, both prior to and after exercise [207]. This suggests that taurine supplementation may exert a beneficial impact on the cardiovascular system in patients with CVD complications. Taurine supplementation in patients (n = 41, 3 g/kg for 30–45 days) who underwent aortocoronary artery bypass (ejection fraction ≤ 40%) showed decreased left ventricular end-diastolic volume with left ventricular dysfunction before revascularization [208]. In addition, patients (n = 17) with congestive heart failure (ejection fraction ≤ 50%) given 3 g/day taurine for 6 weeks showed significant improvement in systolic left ventricular function compared to the control CoenzymeQ10-treated group [209]. Recently, it was shown that blood concentrations of taurine in two siblings (a 6 year old girl and a 15 year old boy), who were diagnosed with a homozygous variant (Gly^399^Val) in the taurine transporter gene, SLC6A6, led to taurine deficiency [19]. Blood taurine was almost undetectable in the two siblings, ranging from 6–7 µmol/L and correlated to the clinical diagnosis mild cardiomyopathy (i.e., left ventricle dysfunction) and visual impairment in both siblings. In fact, continual supplementation of taurine tablets (100 mg/kg/day for 24 months), increased taurine levels to normal levels and corrected cardiomyopathy in both siblings [19], possibly through restoring mitochondrial dysfunction and improving cardiac energy metabolism [23,210]. As the abnormal increase in intracellular [Na^+^]i and overload of [Ca^2+^]i are critical steps in cardiac damage induced via hydrogen peroxide or ischemic reperfusion [211], the mechanisms underlying such long-term or continual taurine supplementation in the above studies may take effect through the inhibition of the Na^+^/Ca^2+^ exchanger, subsequently leading to indirect maintenance of intracellular [Na^+^]i and [Ca^2+^]i homeostasis [1,29,211]. 

It is not clear whether taurine has a beneficial effect in pre-hypertensive patients. Taurine supplementation was evaluated in patients with stage 1 hypertension in a randomized double-blind clinical study [212]. Pre-hypertensive patients were supplemented with 1600 mg/day taurine for 12 weeks, after which blood pressure was significantly decreased compared to placebo, particularly in those with high blood pressure [212]. The mechanism by which taurine decreased blood pressure was via increased hydrogen sulfide synthesizing enzymes and decreased agonist-induced vascular reactivity in mesenteric arteries. This was via taurine being able to inhibit the transient receptor potential channel subtype 3-mediated calcium influx. In another study, a higher dose of taurine for a shorter time (6 g/day for 7 days) decreased blood pressure in patients with hypertension. The mechanisms in these studies were shown to be dependent on the modulation of an overactive sympathetic system [213].

Vasodilation also occurred even after a very large but safe dose of taurine supplementation (5 g/day for 6 days) in a phase 1/2 clinical trial in patients with vascular dysfunction induced by the CVD risk factor, known as hyperhomocysteinemia (caused by cystathionine β synthase deficiency) [214]. Further clinical studies demonstrated that patients with type-2 diabetes following supplementation of taurine (500 mg/3 times/day for 2 weeks) also attenuated vascular dysfunction assessed by lower FMD (flow-mediated dilatation), indicating that taurine alone reverses endothelial dysfunction, which is the primary path to atherosclerosis. However, the other CVD parameters, such as heart rate, ejection rate, or inflammation (detected by hsCRP), were not found in these patients [215], unlike those with heart failure [207]. Supplementing type-2 diabetic patients with 1500 mg/day taurine for 90 days was shown to lower arachidonic acid levels compared to control subjects. Arachidonic acid induces platelet aggregation [216]. In fact, taurine deficiency may be a risk factor for the development of type-2 diabetes, as taurine concentration is significantly lower in diabetic patients (n = 59, 0.2 to 0.9 mmol/L) with the common associated comorbidity, hypertension, compared to healthy subjects (n = 28, 0.5 to 1.2 mmol/L) [217]. As such, taurine supplementation may prevent and/or restore hyperglycemia and its associated comorbidity. Recently, in a double-blind placebo-controlled clinical study in type-2 diabetes patients, it was noted that a high dose of taurine supplementation (n = 45, 3 g/day for 8 weeks) improved fasting blood sugar levels from 169 to 155 mg/dL and reduced triglycerides and low-density lipoprotein cholesterol concentrations [218]. However, at a lower dose for a longer period (1500 mg/day for 3 months), taurine had no effect on fasting blood sugar levels [219]. The serum levels of superoxide dismutase, an antioxidant, was significantly increased, which was associated with decreased inflammation (hsCRP, TNF-α, IL-6) following 3 g/day taurine supplementation for 8 weeks (n = 50) in type-2 diabetic patients compared to control subjects [220].

The higher doses of taurine supplementation may be mediated by decreasing lipid peroxidation and reducing advanced glycation end products, which are produced in patients with type-2 diabetes when the elevated levels of glucose bind to plasma proteins through a process known as glycation reaction, which can lead to oxidative stress [221].

It is clear that long-term taurine supplementation or continual taurine supplementation improves heart function, is anti-hypertensive, shows promise as a treatment in pre-hypertensive patients, and has beneficial effects in patients with type-2 diabetes (Table 1).

## 7. Conclusions

A plethora of evidence reveals the anti-inflammatory properties of taurine. Consumption of seafood abundant in taurine may be a factor in preventing CVD, including CAD, and prolonging life expectancies. Further clinical investigations are required to confirm that foods high in taurine, or taurine supplementation, are a viable adjunct treatment of CVD and its complications in humans.

## Figures and Tables

**Figure 1 nutrients-12-02847-f001:**
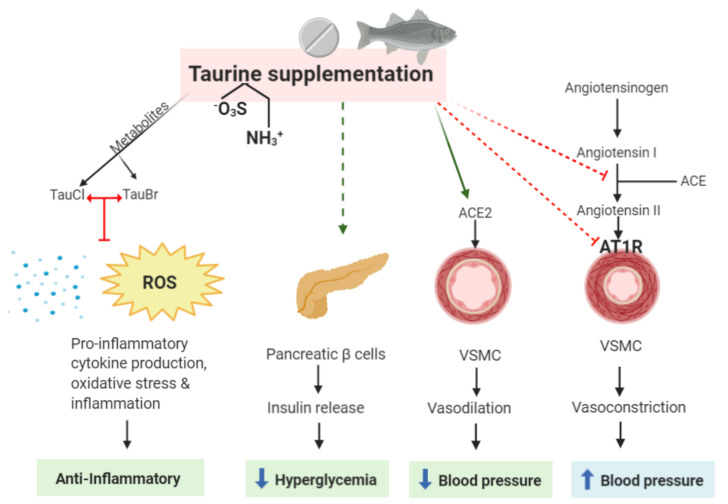
Possible sites of action of taurine as a beneficial agent in treating cardiovascular disease (CVD) and its anti-inflammatory properties. The taurine metabolites, TauCl and TauBr, exert anti-inflammatory properties by blocking the production of pro-inflammatory cytokines, oxidative stress (ROS), and inflammation. Taurine itself may directly reduce hyperglycemia, activate the protective arm of the renin-angiotensin system (RAS) by increasing ACE2 expression, and inhibit the harmful axis of the RAS by reducing Ang II production, which is currently one of the main targets in treating CVD and coronary heart disease (CAD). It must be emphasized that taurine supplementation eliciting insulin release may be detrimental/have no effect in patients who are obese and have already elevated insulin levels. TauCl, taurine-chloramine; TauBr, taurine-bromamine; ROS, reactive oxygen species; VSMC, vascular smooth muscle cell; ACE, angio-converting enzyme; AT1R, AngII type 1 receptor.

**Figure 2 nutrients-12-02847-f002:**
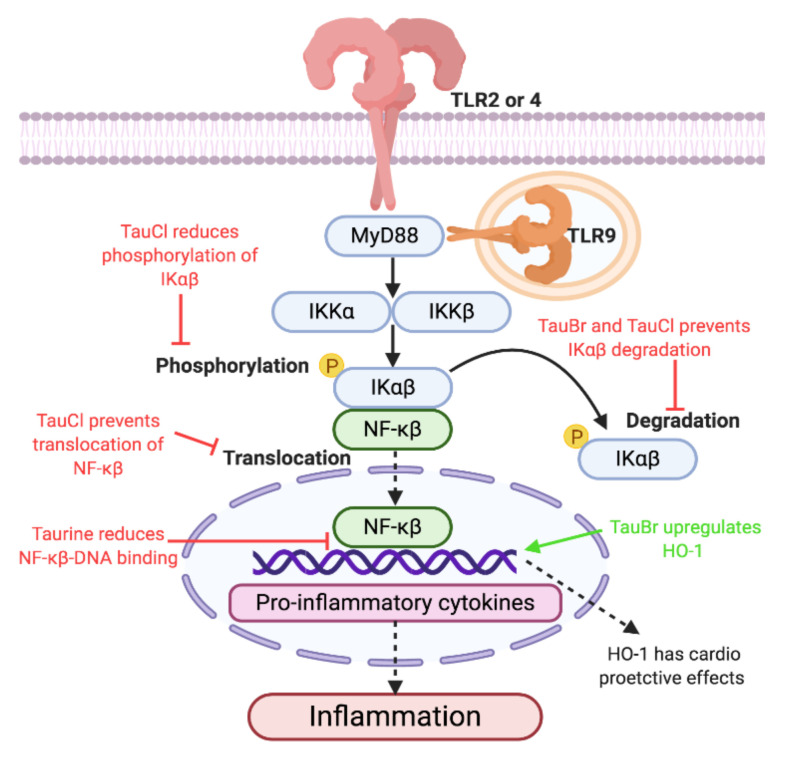
Proposed anti-inflammatory mechanisms of taurine, taurine-bromamine, and taurine-chloramine via TLR/MyD88/NF-Kβ (Toll like receptors/Myeloid differentiation primary response 88/Nuclear factor kappa-light-chain-enhancer of activated β cells) signal transduction. Due to the ability of taurine and its metabolites to reduce inflammation through TLR/MyD88/NF-Kβ signal transduction, using them as therapeutics could result in decreased production of inflammation. Decreasing inflammation could improve cardiovascular disease (CVD) pathologies, as chronic inflammation caused by TLR activation has been implemented in disease pathogenesis and development. Therefore, the anti-inflammatory effect and increased release of heme oxigenase 1 (HO-1) may be therapeutic in CVD pathologies.

**Table 1 nutrients-12-02847-t001:** Highlighted studies on taurine implications for beneficial effects to CVD.

Ref.	Subject	Dose/Time	Outcome
[205,207]	Heart Failure	1500 mg/day, 2 weeks	Improved exercise capacity, reduced TC/HDL-c, and reduced hsCRP
[208]	Chronic Heart Failure	3000 mg/day, 30–45 days	Decreased left ventricular end-diastolic volume with left ventricular dysfunction before revascularization
[209]	Chronic Heart Failure	3000 mg/day, 6 weeks	Improvement in systolic left ventricular function
[19]	Cardiomyopathy	100 mg/day, 24 months	Restored taurine concentration and corrected left ventricular dysfunction
[212]	Stage 1 hypertension	1600 mg/day, 12 weeks	Improved vasodilation and reduced blood pressure
[213]	Hypertension	6000 mg/day, 7 days	Reduced blood pressure
[214]	VascularDysfunction induced by CBS deficiency	5000 mg/day, 6 days	Improved vasodilation
[215]	Type-2 diabetes with vascular dysfunction	1500 mg/day, 2 weeks	Reversed vascular dysfunction
[216]	Type-2 diabetes	1500 mg/day, 90 days	Decreased platelet aggregation
[218]	Type-2 diabetes	3000 mg/day, 8 weeks	Decreased FBS, triglycerides, and LDL-c
[220]	Type-2 diabetes	3000 mg/day, 8 weeks	Increased SOD and decreased hsCRP, TNF-α, and IL-6

CBS, cystathionine β synthase deficiency; FBS, fasting blood sugar; HDL-c, high-density lipid cholesterol; hsCRP, high-sensitivity C-reactive protein; LDL-c, low-density lipid cholesterol; TC, total cholesterol.

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
