# Peer review of "The Anti-Inflammatory Effect of Taurine on Cardiovascular Disease"

_nutrients, 2020, doi:10.3390/nu12092847_

Round 1

Reviewer 1 Report

This article reviews cardio-protective effect of taurine and focuses anti-inflammatory and immunomodulatory actions as possible mechanisms. The manuscript is well-organized as a whole. However, there are some points to be improved.

Is the subtitle “From seafood to supplements” is proper? There are few statements concerning seafood and taurine. If the authors would like to keep it, taurine in seafood should be entered in the manuscript. Considering from the content, it is preferable to use the terms, anti-inflammatory and/or immunomodulatory.

In this review article, the authors point out cardio-protective effects of taurine from inflammation or immunological point of view. It is unprecedented and interesting angle. However, there are few explanations about the relationship between cardiovascular disease and anti-inflammatory or immunomodulatory effects of taurine. In Figure 1, “Anti-inflammatory effects” and “Blood pressure” are separately described. So it is difficult to understand how anti-inflammatory effect of taurine affect the cardiovascular system including blood pressure. Similarly, anti-inflammatory effects of taurine-chloramine and taurine-bromamine are shown in detail in 5.2. and 5.3. However, the explanation concerning how these compounds affect cardiovascular system is insufficient. The authors need to provide more explanation.

As mentioned in page 4 (lines 152-153), diabetes is a risk factor associated with CAD. In addition, obesity is also well-established risk factor responsible for CAD. Obesity is considered chronic low-grade inflammation and an activation of immune system. It is preferable that the authors treat obesity as a risk factor for CAD, because inflammation and immune system are important common factors connecting CAD, diabetes and obesity.

Author Response

Reviewer 1:Comments and Suggestions for Authors

This article reviews cardio-protective effect of taurine and focuses anti-inflammatory and immunomodulatory actions as possible mechanisms. The manuscript is well-organized as a whole. However, there are some points to be improved.

  1. Is the subtitle “From seafood to supplements” is proper? There are few statements concerning seafood and taurine. If the authors would like to keep it, taurine in seafood should be entered in the manuscript. Considering from the content, it is preferable to use the terms, anti-inflammatory and/or immunomodulatory.

New title:  The anti-inflammatory effect of Taurine on cardiovascular disease.

  1. In this review article, the authors point out cardio-protective effects of taurine from inflammation or immunological point of view. It is unprecedented and interesting angle. However, there are few explanations about the relationship between cardiovascular disease and anti-inflammatory or immunomodulatory effects of taurine. In Figure 1, “Anti-inflammatory effects” and “Blood pressure” are separately described. So it is difficult to understand how anti-inflammatory effect of taurine affect the cardiovascular system including blood pressure.

Thankyou for your comment.

The following paragraph was edited for clearer understanding of taurine link with cardiovascular and blood pressure. And explains why blood pressure and anti-inflammatory effects of taurine are incorporated into the Figure 1 separately. 

Line 143-151: In another study, rats were exposed to mechanical stress to elevate blood pressure and supplemented with 200mg/kg/day of taurine [49]. It was found that in the control group ACE was overactivated, contributing to the mechanism of elevated blood pressure [49]. This overactivation of ACE was counterbalanced with ACE2 upregulation after taurine supplementation, and was correlated with a reduction in blood pressure[49]. Therefore, in the early stages of vascular disease such as hypertension, taurine can interact with the beneficial axis of RAS via upregulating ACE2.

  1. Similarly, anti-inflammatory effects of taurine-chloramine and taurine-bromamine are shown in detail in 5.2. and 5.3. However, the explanation concerning how these compounds affect cardiovascular system is insufficient. The authors need to provide more explanation.

Thank you for your constructive comment; we agree with your comment that explanations pertaining to taurine and its metabolites linking with CVD were vague. To rectify this we have provided links between TLR inflammation and CVD in lines 327-335, 374-376, and we have edited 5.2 and 5.3 to further explain the role that TauBr and TauCl should be considered as therapeutics for CVD 339-487.

344-252: Chronic dysregulated TLR activation has been reported in a plethora of non-infectious pathologies and autoimmune diseases, including CVD [84, 85], atherosclerosis [86, 87] and type-2 diabetes [88, 89]. As demonstrated by animal TLR knockout models [90-96] and studies involving TLR deficiency [97, 98] and polymorphisms in humans [99-111], absence or improper function of TLRs can attenuate inflammation, influence disease susceptibility and restore homeostatic balance. Thus, TLRs have become targets of TLR inhibition therapies [112-116]. This review will discuss the potential inhibitory and immunomodulating abilities of taurine and its metabolites on TLR signal transduction, which may make them suitable anti-inflammatory therapeutics in CVD pathologies.

286-288:Due to the anti-inflammatory nature of taurine, it may have cardioprotetcive effects in CVDs, which are described as chronic inflammatory diseases [85, 86].

305-309: TauCl and TauBr have been shown to modulate the TLR/MyD88/NF-K signal transduction in human and murine macrophages. This will be further investigated in the review, focusing on the potential therapeutic abilities of TauCl and TauBr to target inflammation through TLR pathways. The potential benefits these metabolites may have on inflammation and CVD will also be discussed.

311-335: 5.2 Therapeutic aspects of taurine-bromamine

Taurine is the single most abundantly free amino acid found in leukocyte cytosol (20-50nM) [149], which exerts antioxidant effects by scavenging reactive oxygen species (ROS), such as hypohalous acids [150]. Hypobromous acid (HOBr) is a ROS generated by activated neutrophils and eosinophils at sites of inflammation through enzymatic activity of neutrophil myeloperoxidase (MPO) [151] and eosinophil peroxidase [152]. Taurine readily reacts with HOBr to produce the less toxic conjugate taurine bromamine (TauBr) [153]. TauBr protects cells against HOBr damage [154], and has demonstrated anti-microbial [155, 156], -bacterial [157], -inflammatory [158] and -oxidative [158] effects. TauBr has been suggested as a potential topical treatment for propionbacterium granulosum in acne vulgaris, due to its anti-microbial and anti-inflammatory effects. TauBr demonstrated potent anti-bacterial effects similar to that of hypochlorous acid (HOCl) at low concentrations, while remaining cytoprotective [159, 160]. TauBr exerts its anti-inflammatory effects by down-regulating TNF-induced NF-K inflammation by inhibiting oxidation and degradation of nuclear factor of kappa light polypeptide gene enhancer in B-cells inhibitor alpha (IkB) [161]. Additionally, in vitro treatment of TauBr in immune cells and fibrobasts is able to reduce the production of TNF-a, IL-6, IL-10, IL-12p40 [157, 162], and upregulates heme oxygenase-1 (HO-1) [163]. HO-1 is an enzyme that is able to restore homeostatic antioxidant balance during CVD and atherosclerosis [164] The ability of TauBr to associate with and modulate TLR expression and activation has yet to be investigated. However, TLRs 1, 2, 4-10 exert pro-inflammatory modulations through the MyD88-dependnet/NF-K pathway, which utlises IkB for progression of signal transduction [79, 165]. Chronic unregulated activation of TLRs has been demonstrated in vascular pathologies, including CVD [85, 86] and atherosclerosis [87, 88]. Therefore, as TauBr can increase HO-1 and down-regulate NF-K inflammation, its investigation into the ability to attenuate exacerbated and chronic inflammation in CVD and atherosclerosis through TLR signal transduction disruption should be investigated. 

331-396: 5.3. Therapeutic aspects of taurine-chloramine

 HOCl is a ROS, secreted by macrophages, neutrophils and eosinophils during inflammation [166-168]. HOCl acts as an endogenously produced anti-bacterial [169, 170], -fungal [171], -parasitic [119] and –viral agent [119]. However, excessive and unregulated production leads to oxidative stress [172], inflammation [173] and necrosis [174, 175]. Dysregulation of HOCl has been associated with pathologies including atherosclerosis [176-178], acute coronary syndrome [179], obesity [180] and Type II diabetes [181]. Taurine reacts with HOCl to form TauCl [182]. TauCl is a less toxic oxidant that has cytoprotective properties [183], reduces acute [184] and chronic inflammation (caused by HOCl) [185], decreases oxidative stress [186] and is able to cause leukocytes damaged by acute inflammation to undergo apoptosis while preserving healthy cells [187-190] . Unlike TauBr, TauCl has been shown to inhibit mediators of TLR2, 4 and 9 inflammation. Thus, TLRs could be a potential target of TauCl to attenuate chronic inflammatory pathologies. While TauCl is not an activating ligand of TLRs, TauCl treated macrophages with lipoarabinomannan and zymosan (TLR2 actiavting ligands), lipopolysaccharide (TLR4 activating ligand) and CpG OGN (TLR9 activating ligand) demonstrated inhibited production of iNOS, NO and TNF in a dose-dependent manner (maximal efficacy was observed at 0.8mM and resulted in 99% of NO and 48% of TNF) [191, 192]. Furthermore, TauCl was able to supress upregulation of TNF and iNOS mRNA [191, 192] . Therefore, TauCl could have a potential physiological role during protection against tissue injury produced by TLR inflammation. Additionally, TauCl is able to exert anti-inflammatory effects through modulation of NF-K signal transduction and inflammation. TauCl is able to prevent ERK1/2 and NF-K activation [193], and migration of NF-K into the nucleus in TauCl treated rat macrophages [194] and human synovial fibroblasts [182]. TauCl is also able to prevent degradation of IkB by modifying the molecular backbone [182, 194, 195]. Decreased phosphorylation and resistance to degradation of IkB results in decreased activity of IkB kinase required for NF-K signaling [182, 194, 195]. Taken together, TauCl may exert its anti-inflammatory effects by indirectly preventing TLR activation, achieved by preventing molecules necessary for TLR/MyD88/NF-K signal transduction and transcription of pro-inflammatory products. Inflammatory contributions by TLRs have been implicated in the pathogenesis and disease development of CVD diseases [88]. TLR2 and 4 have been shown to be atherogenic, while conflicting research regarding TLR9 in involvement CVD pathologies has been shown in the litertaure [88]. Mice susceptible to CVD with TLR2 deficiency and human in vitro studies investigating atherogenic abilities of TLR have shown the TLR2 plays a role in regulating ROS-induced endothelial and vascular inflammation and [196], plaque formation [197], migration of vascular smooth muscle cells from the tunica media to the intima [198] and influences plaque stability [199]. Comparable results have been demonstrated in studies involving TLR4, as TLR4 has been associated with endothelial and vascular inflammation [200, 201]; atherosclerotic plaque formation [202]; increases macrophage, foam cell and lipid content of plaques [202-205]; hypertension [206], plaque rupture [207]; and migration of vascular smooth muscle cells [208]. Finally, TLR9 has shown to influence plaque formation and macrophage infiltration [209] and endothelial regeneration [210]. TauCl may provide potential therapeutics in CVDs by targeting TLR2, 4, 9/MyD88/NF-K and dampening the inflammatory response (Figure 2).

TLRs have been shown to play a role in chronic inflammatory diseases, including CVD [85, 86] and atherosclerosis[87, 88]. Deficiencies and polymorphisms have shown beneficial results in disease severity and susceptibility [98-112]. Thus, TLRs have become targets of pharmaceuticals and small molecules treatments in TLR inhibition therapies [113-117]. TLRs exert damaging effects through chronic unregulated inflammation through the TLR/MyD88/NF-K pathway [81]. Taurine and its metabolites (TauBr and TauCl) may represent therapeutic amino acids that can indirectly reduce inflammation caused by TLR activation by disrupting normal signal transduction and transcription of pro-inflammatory molecules. Therefore, taurine, TauBr and TauCl could have potential for clinical application in CVD by dampening inflammation and restoring homeostatic balance. However, due to the limiteobd research investigating taurine, TarBr and TauCl interactions with TLRs, and the lack of in vivo models in CVD pathologies more research is required.

We have also created Figure 2 to illustrate the proposed link between taurine and its metabolites on decreasing inflammation through the TLR2,4,9/MyD88/NF-kB pathway.

Figure 2: Proposed anti-inflammatory mechanisms of taurine, taurine-bromamine and taurine chloramine via TLR2, 4, 9/MyD88/NF-K signal transduction. Due to the ability of taurine and its metabolites to reduce inflammation through TLR/MyD88/NF-K signal transduction, using them as therapeutics could result in decreased production of inflammation. Decreasing inflammation could improve CVD pathologies, as chronic inflammation caused by TLR activation has been implemented in disease pathogenesis and development. Therefore, the anti-inflammatory effect and increased release of HO-1 may be therapeutic in CVD pathologies

  1. As mentioned in page 4 (lines 152-153), diabetes is a risk factor associated with CAD. In addition, obesity is also well-established risk factor responsible for CAD. Obesity is considered chronic low-grade inflammation and an activation of immune system. It is preferable that the authors treat obesity as a risk factor for CAD, because inflammation and immune system are important common factors connecting CAD, diabetes and obesity.

Thankyou for this comment,

These changes have been made accordingly. This paragraph now links both Diabetes and Obesity as independent risk factors for CAD and illustrates that taurine supplementation has beneficial effects on both independently. It discusses obesity and its link to inflammation and CAD.

197-199: Therefore, further research is required in diabetic patients to understand the degree in which taurine exerts cytoprotective properties can contribute to the prevention and treatment of type-2 diabetes in larger randomized population groups.

Line 200-221: Those who are considered overweight or obese are at a higher risk of developing type-2 diabetes and insulin resistance, conditions that are associated with accelerated CAD[60]. Obesity is associated with risk factors for CAD due to hyperglycemia/diabetes, high cholesterol, high blood pressure, and metabolic syndrome. Interestingly, obesity itself is an independent risk factor for CAD[61]. Ventricular hypertrophy, diastolic dysfunction and aortic stiffness are all pathologies associated with both functional and structural changes to the heart caused by obesity [62]. Vascular atherosclerotic lesions are far more prevalent and severe in patients with higher BMIs compared to those with normal BMIs, indicating a link between prevalence of atherosclerosis and obesity[63]. Obese patients present with overexpression of pro-inflammatory cytokines including TNF-α and IL-6 and immune cells (phagocytes and macrophages) resulting in elevated inflammation[64]. Taurine has shown to be beneficial in reducing obesity related inflammation. Obese women administered taurine for an eight week period, had increased adiponectin levels and a reduction in inflammation biomarkers (including high-sensitivity C-reactive protein)(hsCRP) and lipid peroxidation (TBAR) [65]. This highlights a role of taurine supplementation in reducing obesity related inflammation, which can reduce the development of CAD. A key target to prevent CAD is through weight loss. Taurine deficiency is associated with promoting obesity in both animal and human studies[66], a cycle that is negated by taurine supplementation[67]. It has been suggested that supplementation with taurine or activation of its synthesis may be a potential therapy to reduce obesity. Following this hypothesis a recent publication in a mouse model of obesity, showed that prolonged taurine supplementation resulted in weight loss suggested to be via adipogenesis inhibition[68]. Given this information supplementation with taurine may be an option to reduce the risk of CVD and CAD in obese patients through reducing BMI.

Reviewer 2: Comments and Suggestions for Authors

  1. This is a review paper dealing with taurine and cardiovascular disease. The subject was recently reviewed (Moussa et al 2020, Liu et al 2020, Bkaily et al 2020, Quizoni et al 2020, etc..) and this review did not add something new to the field of taurine. The review is not well focused and do not take into account the must important aspect of the effect of taurine which is mainly due to its co-transported Na+and its indirect regulation of Ca2+ homeostasis and Ca2+-dependent mechanisms. These aspects are well reviewed by the groups of Schaffer and Bkaily. The latter was completely ignored and the authors should take this into consideration.

Thankyou for your comment. We have now added the most important aspect of the effect of taurine shown in the following paragraphs below.

co-transported Na+ and its indirect regulation of Ca2+ homeostasis

420-424: As the abnormal increase in intracellular [Na+]i and overload of [Ca2+]i are critical steps in cardiac damage induced via hydrogen peroxide or ischemic reperfusion [216], the mechanisms underlying such long-term or continual taurine supplementation in the above studies may take effect through the inhibition of the Na+/Ca2+ exchanger, subsequently leading to indirect maintenance of intracellular [Na+]i and [Ca2+]i homeostasis [1, 29, 216].

Ca2+-dependent mechanisms:

128-148: As AngII could trigger the Na+/Ca2+ exchanger, leading to an influx of intracellular calcium ion ([Ca2+]i), it is possible that taurine may regulate this ion exchanger in cardiac myocytes and inhibit AngII effects on cardiac hypertrophy [53]. Interestingly, AngII-incubation followed by short-term taurine (24 hours) administration in neonatal rat myocyte decreased [Ca2+]i [53]. However, vice-versa incubations had no effect on the AngII-induced increase in [Ca+]i, [53]. As such, it is not clear whether taurine directly inhibits AngII via AT1R, blocks the conversion of AngI into AngII or simply influences the signaling cascade of AngII in cardiac hypertrophy. Bkaily et al (2020), have proposed that short-term taurine supplementation may stimulate Ca2+-dependent mechanisms, whereby, taurine is co-transported with Na+ on cardiomyocytes, subsequently leading to an increase of [Na+]i and [taurine]i [1]. The increase in [Na+]i could signal the Na+/Ca2+ exchanger resulting in elevated [Ca2+]i [1]. Although short-term taurine supplementation may be beneficial, as cardiomyocytes rely on [Ca2+]i for contractility function[1]  , further studies are required to examine the mechanism of taurine effect on AngII in cardiac hypertrophy.

  1. A major criticism is that many sentences had no references which is considered non-ethical. Each sentence and reference to results in the literature should have a reference unless that sentence belongs to the authors.

We have taken your advice and went through the sentences where it requires references.

  1. In many occasions, the authors used very long sentences making it difficult for the reader to follow and understand the message.

We have edited the manuscript again and made sure to shorten the very long sentences.

  1. In addition, some paragraphs have nothing to do with taurine such as in lines 3-38 and 107-121 (dealing with the Ang II system and not taurine), lines 195-211 (dealing with TLRs and inflammation), and lines 266-276 (dealing with HOBr). The authors should omit these paragraphs.

Thankyou for your comment ,we have revised our sentences/paragraphs and believe it builds the rationale to linking taurine.

We have revised lines 3-38, it is our introduction to the manuscript and it is about taurine.

In regard to lines 107-121, we have explained the major regulatory system of the cardiovascular system, renin angiotensin system. We have provided the fundamental understanding of the renin angiotensin system to make the link between taurine and AngII as well as the other renin angiotensin system aspects more clear.

We agree that to lines 195-211we not written concisely. While we maintain that parts of the introduction is needed to explain TLRs, more information regarding taurine and the innate immune system was needed. To rectify this we have extensively edited Section 5, to better correlate TLR, CVD and taurine + its metabolites, and have added original articles discussing taurine deficiency, supplementation and role and the innate immune system.  

232-309:5.1 Taurine, toll-like receptors and cardiovascular disease

The innate immune system is an evolutionary preserved host defense system, responsible for first-line protective mechanisms against infiltrating pathogens [72]. Integral to innate immunity is a class of pattern recognition receptors, referred to as toll-like receptors (TLR) [73]. TLRs are fundamental to innate immunity, as they recognise pathogen-associated molecular patterns (highly conserved, distinct motifs present on pathogens), and respond by exerting robust inflammatory mechanisms in an effort to neurtralise and eliminate pathogenic invasion [74-76]. Furthermore, TLRs are able to respond to danger-associated molecular patterns (host-derived endogenous alarmins), released by damaged, dying or necrotic cells, independent of pathogenic infection [77]. The end product of sterile inflammation, either through the myeloid-differentiation factor-88 (MyD88)-dependent pathway (TLR1, 2, 4-10) [78] or the MyD88-independent pathway (TLR3 and 4) [79], is ubiquitous among TLRs [80]. However, they display diversity and specialty, as each TLR is activated by unique and specific ligands [81-83]. Chronic dysregulated TLR activation has been reported in a plethora of non-infectious pathologies and autoimmune diseases, including CVD [84, 85], atherosclerosis [86, 87] and type-2 diabetes [88, 89]. As demonstrated by animal TLR knockout models [90-96] and studies involving TLR deficiency [97, 98] and polymorphisms in humans [99-111], absence or improper function of TLRs can attenuate inflammation, influence disease susceptibility and restore homeostatic balance. Thus, TLRs have become targets of TLR inhibition therapies [112-116]. This review will discuss the potential inhibitory and immunomodulating abilities of taurine and its metabolites on TLR signal transduction, which may make them suitable anti-inflammatory therapeutics in CVD pathologies.

Taurine and TauT localization has been reported in cells of the innate immune system, including lymphocytes [117]; neutrophils [118]; phagocytes [119]; and leukocytes [120]. While taurine does not directly associate with TLRs, it has been suggested that taurine may: (i) act as an antioxidant during inflammation induced by oxidative stress [121, 122], (ii) exert cytoprotective [123, 124], anti-inflammatory [122] and anti-neoplastic [125] effects, and (iii) maintain homeostasis during acute and chronic inflammation [118]. Furthermore, pre-treatment with taurine in rat models of Streptococcus uberis mastitis was able to significantly decrease mRNA expression of TLR2, nuclear factor-kappa beta (NF-K) and NF-K DNA binding activity [126] (Figure 2), reducing inducible nitric oxide (iNOS) and TNF concentrations. Interestingly, TLR4 mRNA expression was significantly increased by taurine supplementation [126], suggesting an in vivo TLR-taurine immunomodulating relationship. Therefore, further studies into the direct relationship between taurine and TLRs are warranted.

               The protective and anti-inflammatory effect of taurine has been observed in rodent models of acute lung injury (induced by bleomycin, endotoxin and cigarette smoke exposure), central nervous system injury and chemotherapy toxicities. Rodents treated with taurine prior to and after bleomycin-induced acute lung injury showed an absence of pulmonary fibrosis and reduced inflammation markers including iNOS, intracellular adhesion molecule, TNF, NF-K and interleukin (IL) 2, when compared with the untreated group [127-131]. Similar anti-inflammatory effects of taurine have been replicated in endotoxin [132, 133] and cigarette smoke [134] models of acute lung injury. After spinal cord injuries, treatment with taurine was able to reduce inflammation, decrease accumulation of neutrophils and secondary degenerative consequences in gray matter, attenuate motor disturbances and promote axonal regeneration [135, 136]. Comparable protective anti-inflammatory results were also observed in traumatic brain injuries, as treatment with taurine was able to significantly reduce pro-inflammatory cytokines (including: IL-1; -1; -4; -5; -6; -10; -12p70; 13; 17; TNF; eotaxin; granulocyte colony-stimulating factor; granulocyte-macrophage colony-stimulating factor; interferon- γ; monocyte chemotactic protein-1; and vascular endothelial growth factor) and alleviated severity of the injury [137, 138]. Finally, taurine supplementation has shown to protect cardiac [139], renal [140] [141, 142] and gonadal [143] tissue from cisplatin-induced toxicities, by reducing inflammation caused by oxidative and endoplasmic reticulum stress. Interestingly, while taurine supplementation shows anti-inflammatory effects, TauT deletion shows increased susceptibility [144] and loss of self-healing [145] to parasitic Malaria infection. This suggests that taurine may play a role in launching innate immune system protective inflammatory responses against pathogen infection or taurine may be able to arrest growth and development during sexual and asexual stages [144]. Due to the anti-inflammatory nature of taurine, it may have cardioprotective effects in CVDs, which are described as chronic inflammatory diseases [84, 85].

Prolonged taurine deficiency in animal models has resulted in immune system dysfunction and abnormalities. Feline models fed a taurine-free diet developed leukopenia, a shift in polymorphonuclear and mononuclear leukocytes percentages, increased total monoclear leukocyte count, alterations in sedimentation of white blood cells and augmented concentration of serum γ-globulin [146]. Functional studies performed on polymorphonculear cells demonstrated significantly lower numbers of neutrophils in the respiratory tract and a reduction in phagocytosis of Staphylococcus epidermidis in felines fed a taurine-free diet [146]. Furthermore, lymph node and spleen histological examination demonstrated regressed follicular centres with depleted fibroblasts (reticular cells), mature and immature B lymphocytes and mild extravascular haemolysis [146]. To date, the direct correlation between taurine deficiency and immune consequences in humans is elusive. However, it is suggested that TauT expression may be upregulated during inflammation [147]. Furthermore, the anti-inflammatory properties of taurine may be mediated through regulation of TauT transporter in leukocytes [122]. Following anti-depressant treatment, diminished TauT expression was observed in lymphocytes, resulting in augmented extracellular taurine concentration [122]. The efflux of taurine may account for protection against oxidative stress and inhibition of pro-inflammatory cytokine-mediated damage by enabling an increase in its metabolites taurine-bromamine (TauBr) and taurine-chloramine (TauCl) [122]. TauCl and TauBr have been shown to modulate the TLR/MyD88/NF-K signal transduction in human and murine macrophages. This will be further investigated in the review, focusing on the potential therapeutic abilities of TauCl and TauBr to target inflammation through TLR pathways. The potential benefits these metabolites may have on inflammation and CVD will also be discussed.

The authors insist on the benefits of increasing taurine consumption and seafood. It is well known that it is not the quantity of taurine that is important but the continual supplementation of this non-essential amino acid. This is well reviewed in the literature.

Thankyou for clarifying this aspect of our review, we have now added the following to the manuscript:

In line 463-465 ‘continual’.

It is clear that long-term taurine supplementation and continual taurine supplementation improves heart function, is anti-hypertensive, shows promise as a treatment in pre-hypertensive patients and has beneficial effects in patients with type-2 diabetes (Table 1).’

  1. In lines 77-79, the authors cited that taurine opens an unspecified K+ The K+channel is well specified and was reported to be a delayed outward K+ current (Bkaily et al, 2020 and references within). This should be taken into account.

Thankyou for your insight, we have adjusted our sentences .

78-81: When taurine was administered to thoracic aortae harvested from male Wistar rats, vasorelaxation was induced in a dose-dependent manner via the opening of an unspecified potassium channel [27] possibly the delayed outward K¬+ channel current [1],   as impaired vasorelaxation is one of the main factors leading to CVDs [28]. aurine causes a direct effect on blood vessel function, when administered to arteries in in vitro studies [28].

  1. In section 3.1, the authors mentioned that taurine directly affects the RAS. However, it is well known that taurine reverses the effect of Ang II via decreasing intracellular Ca2+which is reported to be increased by Ang II. This should be considered in the review. It is also recommended to avoid interpretation of the literature.  

Thankyou for your comment. This is a very good point also raised by Review 1. Please refer to above, Reviewer 1’s comment, for the changes we have made. Thankyou

  1. Figure 1 may cause confusion to the reader since it shows that taurine induces the release of insulin which reduces hyperglycemia. The authors should emphasize that, in a normal subject, this will promote hyperinsulinemia leading to hypoglycemia. It should be noted that in obese patients, insulin levels are very high and they suffer from cardiovascular disease.

Thankyou for your comment, we have noted this in our manuscript under the figure1 description.

‘It must be emphasized that taurine supplementation-eliciting insulin release may be detrimental/no effect in patients who are obese and have already elevated insulin levels.

  1. In the same figure, the authors drew a red line indicating that taurine will block Ang II. This has absolutely no sense since taurine will block the physiological effects of Ang II and not Ang II itself or it will block the conversion of Ang I to Ang II. As shown in this particular part of the figure, the reader will understand that blockade of Ang II (synthesis? Physiological effect?) would induce an increase in blood pressure. This should be corrected because, as it stands, it is misleading even to experts in the field.

Thankyou for clarifying and we agree. We have incorporated your advice into our figure1.

Figure 1. Possible sites of action of taurine as a beneficial agent in treating CVD and its anti-inflammatory properties. The taurine metabolites, TauCl and TauBr exert anti-inflammatory properties by blocking the production of pro-inflammatory cytokines, oxidative stress (ROS) and inflammation. Taurine itself may directly reduce hyperglycemia, activate the protective arm of RAS by increasing ACE2 expression and inhibit the harmful axis of RAS by reducing Ang II production which is currently one of the main targets in treating CVD and CAD. It must be emphasized that taurine supplementation-eliciting insulin release may be detrimental/no effect in patients who are obese and have already elevated insulin levels. TauCl, taurine-chloramine; TauBr, taurine-bromamine; ROS, reactive oxygen species; VSMC, vascular smooth muscle cell, ACE, angioconverting enzyme; AT1R, AngII type 1 receptor; and RAS, renin-angiotensin-system.

  1. In lines 183-184, what is the reference of the previous animal studies?

Thankyou for your comment, we omitted this sentence.

  1. It will be worth adding few figures or schematics that may help the reader to better appreciate the review paper.

Thank you for your comment. We do agree that due to complex text complementary figures are needed. We have added Figure 2 to better explain the proposed mechanism by which taurine and its metabolite could be a therapeutic option for reducing TLR inflammation in CVD.

Reviewer 2 Report

This is a review paper dealing with taurine and cardiovascular disease. The subject was recently reviewed (Moussa et al 2020, Liu et al 2020, Bkaily et al 2020, Quizoni et al 2020, etc..) and this review did not add something new to the field of taurine. The review is not well focused and do not take into account the must important aspect of the effect of taurine which is mainly due to its co-transported Na+ and its indirect regulation of Ca2+ homeostasis and Ca2+-dependent mechanisms. These aspects are well reviewed by the groups of Schaffer and Bkaily. The latter was completely ignored and the authors should take this into consideration. A major criticism is that many sentences had no references which is considered non-ethical. Each sentence and reference to results in the literature should have a reference unless that sentence belongs to the authors. In many occasions, the authors used very long sentences making it difficult for the reader to follow and understand the message. In addition, some paragraphs have nothing to do with taurine such as in lines 3-38 and 107-121 (dealing with the Ang II system and not taurine), lines 195-211 (dealing with TLRs and inflammation), and lines 266-276 (dealing with HOBr). The authors should omit these paragraphs.

The authors insist on the benefits of increasing taurine consumption and seafood. It is well known that it is not the quantity of taurine that is important but the continual supplementation of this non-essential amino acid. This is well reviewed in the literature. In lines 77-79, the authors cited that taurine opens an unspecified K+ channel. The K+ channel is well specified and was reported to be a delayed outward K+ current (Bkaily et al, 2020 and references within). This should be taken into account.

In section 3.1, the authors mentioned that taurine directly affects the RAS. However, it is well known that taurine reverses the effect of Ang II via decreasing intracellular Ca2+ which is reported to be increased by Ang II. This should be considered in the review. It is also recommended to avoid interpretation of the literature.  

Figure 1 may cause confusion to the reader since it shows that taurine induces the release of insulin which reduces hyperglycemia. The authors should emphasize that, in a normal subject, this will promote hyperinsulinemia leading to hypoglycemia. It should be noted that in obese patients, insulin levels are very high and they suffer from cardiovascular disease. In the same figure, the authors drew a red line indicating that taurine will block Ang II. This has absolutely no sense since taurine will block the physiological effects of Ang II and not Ang II itself or it will block the conversion of Ang I to Ang II. As shown in this particular part of the figure, the reader will understand that blockade of Ang II (synthesis? Physiological effect?) would induce an increase in blood pressure. This should be corrected because, as it stands, it is misleading even to experts in the field.

In lines 183-184, what is the reference of the previous animal studies?

It will be worth adding few figures or schematics that may help the reader to better appreciate the review paper.

Author Response

(The authors gave the same response as above.)

Round 2

Reviewer 2 Report

The revised manuscript highly improves.

This manuscript is a resubmission of an earlier submission. The following is a list of the peer review reports and author responses from that submission.

Round 1

Reviewer 1 Report

In this review article, Qaradakhi et al. summarize the proposed health effects of the non-proteinogenic amino acid taurine in the context of a variety of physiological systems and diseases. Specifically, they discuss suggested roles in the innate immune response, diabetes, the renin angiotensin system, apoptosis and kidney function. Overall, the authors provide a broad, well-structured overview that will be of interest to the readers of Nutrients. There are a few short-comings that should be addressed to further improve the manuscript.

Concerns that require major changes:

1) There are many orthographical errors throughout the manuscript and the sentence structure is often awkward, which can make it difficult to discern the authors’ meaning at times. There are at least two sentences that are missing a predicate. The authors are therefore encouraged to consult a professional editing service to help them correct this issue.

2) At times, chapter 2 relies too heavily on other reviews instead of citing the primary literature, even on matters directly pertinent to the focus of the manuscript. The authors should select the most pertinent research articles and cite them. This is particularly true for the section from lines 108-121, a lengthy paragraph on the role of taurine in the immune system that relies entirely on two other review articles.

Concerns that require minor changes:

3) It would be interesting to include a brief discussion of how HOCl is conjugated to taurine to yield TauCl, and how this process is regulated (as is provided for TauBr). Similarly, readers may be interested to read a few sentences about how taurine levels are regulated throughout the body beyond food intake.

4) Unclear logic: “In addition, the importance of beta cell function in type-1 diabetes was shown by L’Amoreaux et al. 2010 where they demonstrated that the uptake of taurine in pancreatic cell lines lead to the alteration of the electrical potential of the beta cells, resulting in decreased intracellular insulin levels [64]. Further, studies are required to determine whether this mechanism may also be a potential therapeutic advance for type-2 diabetes by causing a decrease in plasma glucose levels.” (lines 227-232) Why would a treatment that decreases insulin production lead to a decrease in plasma glucose levels? Or were the intracellular insulin levels decreased due to enhanced secretion? Either way, the authors should clarify this point.

5) Lines 248-249: “Although the use of taurine in the therapy against diabetes is elusive there is substantial research investigating the effects of taurine on apoptosis in an extensive display of different cell lines.” It is not clear how the effects on apoptosis are related to the chapter on diabetes. The chapter on taurine and apoptosis does not immediately follow this section.

6) Line 262: MrgD is officially named Mas-related G-protein coupled receptor member D.

7) Chapter 6 might fit better after chapter 4, as the kidneys are functionally somewhat related to the renal angiotensin system, whereas apoptosis is not.

8) Line 479: “Taurine metabolites may either activate TLR […]” This is not discussed in the text and runs counter to the anti-inflammatory properties that are described. If this has been reported, the authors should address it in the main text and discuss how this observation can be reconciled with the apparently intracellular TLR-inhibiting actions of taurine metabolites.

Author Response

For responses to reviewers please see attacheded word document.

Reviewer 2 Report

This review by Quaradaki et al. details the multifactorial effects of taurine. The manuscript is well written but the organization probably needs to be modified. The following are some points for authors to consider:

  1. The introduction section to taurine should also include details on its metabolism/kinetics
  2. Authors have explained in detail about innate immune response in general, there are several resources available for that. Authors can delve in detail on the innate and adaptive immune responses affected by taurine specifically.
  3. Immediately following the section on taurine and immunity authors have included sections on therapeutic effects of taurine chloramine etc. followed by other physiological effects like apoptosis etc. All the immune/physiological responses should be detailed first followed by how it is affected in diseases and finally the role of taurine as therapeutics. This would perhaps make the review easier to read and comprehend.
  4. The physiological effects of taurine also needs section on its role in energy metabolism as CVD, diabetes, cancer all have metabolic components.
  5. Role of taurine in osmoregulation would be pertinent as for eg Ca+ imbalance plays role in CVD like myocardial infarction
  6. Separate sections for cancer/renal etc are included but why was CVD left out when in the introduction it is included/
  7. Taurine is also involved in neuromodulation authors could perhaps briefly include that as this is a general review on the multifactorial effects of taurine.

Author Response

For responses to reviewers, please see attached document.

Round 2

Reviewer 2 Report

The authors have addressed all the major concerns raised.